# Improving UV Curing in Organosolv Lignin-Containing Photopolymers for Stereolithography by Reduction and Acylation

**DOI:** 10.3390/polym13203473

**Published:** 2021-10-10

**Authors:** Jordan T. Sutton, Kalavathy Rajan, David P. Harper, Stephen C. Chmely

**Affiliations:** 1Center for Renewable Carbon, The University of Tennessee Institute of Agriculture, Knoxville, TN 37996, USA; jsutton4@gmail.com (J.T.S.); krajan@utk.edu (K.R.); 2Department of Materials Science and Engineering, The University of Tennessee Knoxville, Knoxville, TN 37996, USA; 3Department of Biosystems Engineering and Soil Science, The University of Tennessee Institute of Agriculture, Knoxville, TN 37996, USA; 4Department of Ag & Bio Engineering, Penn State University, University Park, PA 16802, USA

**Keywords:** lignin, 3D printing, stereolithography, additive manufacturing, resin

## Abstract

Despite recent successes in incorporating lignin into photoactive resins, lignin photo-properties can be detrimental to its application in UV-curable photopolymers, especially in specialized engineered resins for use in stereolithography printing. We report on chemical modification techniques employed to reduce UV absorption by lignin and the resulting mechanical, thermal, and cure properties of these modified lignin materials. Lignin was modified using reduction and acylation reactions and incorporated into a 3D printable resin formulation. UV–Vis absorption at the 3D printing range of 405 nm was reduced in all modified lignins compared to the unmodified sample by 25% to ≥ 60%. Resins made with the modified lignins showed an increase in stiffness and strength with lower thermal stability. Studying these techniques is an important step in developing lignin for use in UV-curing applications and further the effort to valorize lignin towards commercial use.

## 1. Introduction

With the global emphasis on environmentally and economically secure industries, the search for high-value lignin-containing products continues at a quick pace. The core target products include fuels, chemicals, and polymers, and much work has been done to study lignin applicability in these areas [1]. Often these studies leverage intrinsic properties of lignin in targeting which products have the most promise, such as high phenolic content to replace the petroleum-derived compounds in adhesives [2], redox behavior of lignosulfonates as electrolytes in flow batteries [3], and relatively high thermal degradation temperature to provide thermal protection to polymer composites [4]. In addition, research has been directed toward the photoactivity of lignin to design UV-resistant polymers [5,6], and lignin has even been studied as a UV blocker for broad-spectrum sunscreen formulations [7,8,9]. Perhaps unsurprisingly, developing lignin-containing photopolymers has attracted much less attention, probably due to the perceived hurdle of lignin blocking light in the UVA and UVB ranges, which are common curing ranges for photopolymers. 

Lignin is the second most abundant natural polymer beside cellulose, and it is the most abundant renewable aromatic feedstock [10]. As such, it has high potential for application in the chemical and pharmaceutical industries. Lignin depolymerization is a topic of growing interest which encompasses isolating and using high-purity, low-molecular-weight products [7]. However, most of the lignin available today is supplied by the Kraft pulping process as a byproduct of cellulose production, and most Kraft lignin is sulfonated and generally requires purification before use [11,12]. Higher-quality lignin can be obtained through other extraction techniques such as any of the several solvent-based fractionation processes [13,14]. These also offer the advantage of being more environmentally friendly through a reduction in hazardous waste [15]. However, with any extraction process, the structure of the resulting lignin will be altered from its natural form [16]. 

Lignin is composed of phenylpropane groups arranged in complex inhomogeneous network structures that vary based on the plant species. Like any polymer, these structures give lignin its unique properties; however, the structural heterogeneity of technical lignins render elucidating its structure–property relationships extremely challenging. For this reason, applying lignin to a product for a specific desired result remains a grand challenge of 21st-century biorefining. Nevertheless, these rich, complex structures also provide the possibility for chemical modifications that can tune its properties for more efficient use, and the synthesis of new chemical active sites on lignin is a topic that has been studied for many years [17]. Modifications including phenolation, esterification, alkylation, and ethylene grafting have been previously reported to improve lignin reactivity, and homogenization, as well as the mechanical performance of lignin-based composites [18,19]. Overall, such chemical modifications are useful for improving lignin compatibility with polymers and polymer blends.

Photopolymer materials are commonly used in adhesives, sealants, and coatings, and an emerging application of photopolymers that has significant economic impact is in additive manufacturing (AM) [20]. Stereolithography (SLA) is an AM process utilizing a wide variety of photopolymers whose material properties and requirements have been recently reviewed elsewhere [21,22]. Generally, acrylic and epoxy resins are used in SLA printers in combination with polycarbonates, polyether, polyesters [21], ceramic [23], carbon nanotubes [24], cellulose nanoparticles [25], etc., to achieve tunability of thermo-mechanical properties, visual clarity, and biocompatibility. Despite the challenges associated with lignin photopolymers, lignin-containing SLA resins could offer an environmentally friendly option in a group of petroleum-derived, generally toxic materials [26]. Lignin has been used in a variety of 3D printing applications [27], although its use in stereolithography is limited [28,29]. Our group has previously shown that lignin can be used in commercially available SLA products to generate 3D printable lignin materials [30]. In fact, in applications where lignin is chemically modified, a greater concentration of lignin can be used in the photoactive resins than in those where lignin is merely added as a filler (up to 15 wt.% compared to 1 wt.% for unmodified lignin) [27]. However, the reduced UV transparency imparted by lignin is a barrier to printing with increasing lignin loading amounts. 

Accordingly, we report here our efforts to modify an organosolv lignin to reduce its inhibitive effect in photopolymer cure properties. We chose softwood pine lignin because it is an industrially relevant biorefinery energy crop in the United States [31]. We employed chemical modifications to introduce methacrylate or acrylate moieties onto the lignin macromolecule. In addition, we used chemical reduction with NaBH_4_ to decrease photoactive moieties and enhance the photochemical reactions occurring in the presence of lignin that are relevant to SLA. We then blended the modified lignin in a base resin formulation to produce an SLA photopolymer resin. We performed characterization of the spectroscopic, mechanical, and thermal properties of all the samples. Finally, we used a commercial desktop 3D printer to print with these resins and evaluate print quality.

## 2. Materials and Methods

### 2.1. Reagents and Materials

Lignin was isolated from southern yellow pine (*Pinus* spp.) using an ethanol organosolv technique reported previously [32]. Methacrylic anhydride (Alfa Aesar; Haverhill, MA, USA), acrylic anhydride (Accela; San Diego, CA, USA), sodium borohydride (Acros Organics; Waltham, MA, USA), sodium bicarbonate (Fisher Chemical; Waltham, MA, USA), and dimethylamino pyridine (DMAP, Acros Organics) were used as received. Solvents, including N,N-dimethylformamide (DMF, Fisher Chemical), dimethyl sulfoxide (DMSO, Sigma-Aldrich; St. Louis, MO, USA) and tetrahydrofuran (THF, Fisher Chemical), were used during modification and characterization of lignin. An aliphatic urethane triacrylate (Ebecryl 284 N, Allnex Netherlands BV) was used as the resin oligomer base. The diacrylic monomer 1,6-hexanediol diacrylate (HDODA, UBC Chemicals; Instanbul; Turkey) was used as the reactive diluent. The photopackage components consist of diphenyl(2,4,6-trimethylbenzoyl)phosphine oxide (PL-TPO, Esstech Inc.; Essington, PA; USA) as the photoinitiator and titanium dioxide (white, Harwick Standard; Akron, OH; USA) and pyrolyzed lignin (black) [33] as pigments. Commercial resins Rigid, Grey Pro, and Clear V2 were obtained from Formlabs (Somerville, MA, USA) for determining print parameters. 2-chloro-4,4,5,5-tetramethyl-1,3,2-dioxaphospholane (TMDP, Santa Cruz Biotechnology, Inc.: Dallas, TX, USA) was stored in a vacuum desiccator to protect it from ambient moisture. Endo-*N*-hydroxy-5-norbornene-2,3-dicarboximide (Sigma-Aldrich) was used as received.

### 2.2. Lignin Reduction

Lignin was reduced by reacting it with equal parts sodium borohydride in methanol for 2 h at ambient temperature. The resulting suspension was concentrated using a rotary evaporator, and the afforded solid was washed with deionized water over qualitative filter paper. The resulting lignin samples were dried under reduced pressure at 80 °C for 18 h. 

### 2.3. Lignin Acylation

We used a procedure for lignin acylation that we have reported previously [30,34], which is an adaptation of the method for syringyl methacrylate synthesis reported by Epps et al. [35]. Briefly, one equivalent of either methacrylic anhydride or acrylic anhydride was mixed with lignin (1 equiv. of the lignin OH groups as measured by ^31^P NMR spectroscopy) in the presence of a catalytic amount of DMAP (0.038 equiv. of anhydride) in DMF. The acylation proceeded for 48 h at 60 °C. Acylated lignin was recovered from the solution by quenching with a saturated solution of sodium bicarbonate. The precipitated lignin was recovered, washed with water until the pH of the washes measured neutral, and dried under reduced pressure for 48 h at 50 °C. 

### 2.4. Lignin Characterization

All of the lignin samples were thoroughly characterized before and after each of the chemical modifications. To measure OH group content, lignin samples were phosphitylated using TMDP, and ^31^P NMR spectra were collected using a Varian 400-MR spectrometer (Varian Inc., Palo Alto, CA, USA) operating at 161.92 MHz and 25 °C [36]. NMR spectra were quantified using endo-*N*-hydroxy-5-norbornene-2,3-dicarboximide as an internal standard, and the spectra were referenced to the water-derived complex of TMDP (δ 132.2 ppm). 

FTIR spectra (4000–600 cm^−1^) were collected using a UATR Spectrum Two instrument (PerkinElmer, Llantrisant, UK); samples finely ground with a mortar and pestle were used. The powdered samples were placed in direct contact with the ATR diamond crystal and analyzed using 16 scans per spectrum and 4 cm^−1^ resolution. The FTIR peaks were assigned based on previous reports [37,38,39]. Samples were prepared for UV–Vis measurements by dissolution of lignin in dimethyl sulfoxide (DMSO) at a concentration of 1.0 g/L. Spectra were collected over the range of 200–800 nm using a Genesys 10s UV–Vis spectrophotometer (Thermo Fisher, Waltham, MA, USA). 

Lignin molecular weight (MW) analyses were performed using a Tosoh EcoSEC gel permeation chromatography (GPC) system equipped with a refractive index (RI) detector. Samples were prepared in a 0.75 g lignin/1 mL THF solution with no additional pretreatment required due to the presence of acrylate and methacrylate groups on lignin. Measurements were calibrated using PMMA (500–1.0 × 10^6^ Da) and collected at 25 °C.

### 2.5. Resin Formulation

A base resin was formulated to match closely in mechanical and cure properties to the commercial resins obtained from Formlabs. The base resin is composed of 40% reactive diluent (HDODA), 59% resin base oligomer (Ebecryl 284 N), and 1% photopackage. The photopackage consisted of 0.84% by total resin weight of photoinitiator (PL-TPO) and 0.16% by total resin weight pigment, equal parts black and white (lignin carbon and TiO_22_ respectively). Instead of pigment, lignin-containing resins were composed of additional photoinitiator and substituted 5% lignin in place of oligomer. Resin components were mixed in polypropylene jars. Lignin, photoinitiator, and pigments were dissolved in the reactive diluent for 24 h before mixing in the oligomer to ensure full incorporation. The components were premixed in a kinetic mixer (FlackTek, Speedmixer Dac 150 FVZ) at 1000 rpm for 1 min as a wetting step and then mixed again at 1500 rpm for 1 min. All complete resin mixtures were stored in sealed opaque containers to avoid unwanted reaction in ambient light.

Uncured resins were measured for viscosity and cure properties. Viscosity was measured using a TA Instruments AR-G2 rheometer. The instrument was configured in a cone and plate geometry with a 40 mm diameter cone at a truncation gap of 56 microns. Shear rate was measured over a range of 0.1 to 1000 s^−1^ in ambient lab conditions.

Resin cure properties were characterized by the print parameters penetration depth (Dp) and critical cure dosage (Ec) to evaluate printability. The penetration depth is defined at the depth at which irradiance is reduced to 1/e the level at the surface. Critical cure dosage is defined as the exposure corresponding to the transition from liquid phase to solid phase at the gel point. The method for determining these parameters was first established by Jacobs [40] and is described more fully in our previous study [30]. Briefly, squares are cured on a quartz glass substrate at various dosages and then measured for thickness. Cure thickness and the log of dosage are linearly related, where the slope of the line relates to the penetration depth and the intercept relates to the critical cure dosage.

### 2.6. Cure Scheme and Material Characterization

Resins were cast between 15 by 15 cm^2^ quartz glass sheets separated at 1 mm. A controlled cure scheme was chosen to avoid cracking and voids incurred by excessive shrinking. The resins were cured for 20 min in a 10 W 395–400 nm UV cure chamber and then transferred to a 400 W metal halide UV lamp (Uvitron International Inc., West Springfield, MA, USA) for 1 h. The resin sheets were inverted halfway through each step. Resin sheets were removed from the glass and laser cut to tensile bars according to ASTM D638 type V dimension. 

Ultimate tensile strength, Young’s modulus, and elongation at fracture were determined for all resins on an Instron 5567 dual column universal testing machine using a 500 N static load cell. All tests were performed according to ASTM D638 standard procedure. A 1 mm/min extension rate was used for all samples, as well as a preload force of 5 N. All samples were sanded, cleaned, and evaluated prior to testing for voids, cracks, or any other sources of stress concentration.

## 3. Results and Discussion

Acrylate, methacrylate, and fumarate resins are commonly used in commercial SLA applications, and in recent years inclusion of bio-based materials such as vanillin, cellulose, caprolactone, lactic acid, alginate, globular proteins, and soybean oil have been investigated [41]. We have previously shown that lignin can be used to make working SLA resins in commercial printer and resin systems [30]. However, at higher lignin loadings, UV penetration depth through the resins decreases significantly, requiring higher cure dosages and limiting the potential lignin content based on the capabilities of the printer. The reduction in UV penetration can be attributed to light absorption by chromophoric systems in lignin [42]. Reductions using NaBH_4_ have been shown to disrupt these chromophores and decrease absorption, especially in the wavelengths near 405 nm, which are pertinent to common photoinitiators used in 3D printing. Accordingly, we chose to reduce lignin with NaBH_4_ as a strategy to improve cure properties in lignin resins. In addition, these reductions have the added benefit of affording additional –OH groups in lignin, which we transformed to append polymerizable acrylates (Figure 1). Similarly, acrylic anhydride was chosen as an alternate acylation method due to faster cure conversions in polyacrylates versus polymethacrylates [43]. Additionally, since acrylates and methacrylates are the most common bases for SLA resins, these chemical modifications will enhance the compatibility of our lignin with commercial formulations.

### 3.1. Characterization of Reduced and Acylated Lignin

Each type of modified lignin was characterized using ATR-FTIR, ^31^P NMR, and UV–Vis spectroscopies. Henceforth, these lignin samples will be labeled as Pine-M for methacrylic anhydride acylation, Pine-A for acrylic anhydride acylation, Pine-R for sodium borohydride reduction, and Pine-MR for reduction followed by methacrylic anhydride acylation. The FTIR spectra in Figure 1 show an increase in peak signal corresponding to C=O stretching, C–O–C stretching, and –CH_2_ bending vibrations for acylated lignin samples (Pine-MR, Pine-A, and Pine-M) compared to the other samples. Likewise, these samples see a reduction in –OH functional groups. This confirms the success of the acylation reaction at the –OH sites.

We also used ^31^P NMR spectroscopy to identify important changes to the structure of lignin. The results of these measurements are tabulated in Table 1. As expected, we detect a decrease in –OH groups among Pine-M, Pine-A, and Pine-MR, which indicates that the acylation reaction was successful. In addition, we detected a substantial increase in aliphatic –OH groups in Pine-R as compared to the unmodified lignin, which indicates NaBH_4_ can reduce aliphatic carbonyls in lignin to afford aliphatic –OH groups. These carbonyl groups have been implicated as chromophores that increase light absorption by lignin around 400 nm [42]. 

We also collected UV–Vis absorption data in triplicate for lignin samples before and after each modification. These data indicate a significant decrease in absorption in the 380–450 nm range. At 405 nm, absorption decreased in Pine-M, Pine-A, and Pine-R by approximately 20–30%, while absorption decreased by greater than 60% in Pine-MR samples compared to the unmodified lignin (Figure 2). 

We chose chemical reduction as a strategy to decrease the number of chromophores in lignin. While the absorption curves show that this effect was achieved, acylation also evidently decreases absorption as well. While the exact reason for this change is unclear, we attribute it to an alteration in the electronic structure of lignin. In this way, acylation serves two purposes in lignin photopolymers: it is a means to append polymerizable groups to lignin and to also decrease the UV blocking effect imparted by unmodified lignin. Furthermore, it is perhaps unsurprising that the combination of acylation and chemical reduction improved the UV transparency more than any single modification technique since chemical reduction affords additional reactive sites for acylation. 

We performed size-exclusion chromatography on the lignin samples to observe the effect the reduction reaction had on molecular weight (MW). We note a negligible change in MW in reduced versus non-reduced samples. Pine-M has a number-average molecular weight M_n_ = 1279 g/mol and a weight-average molecular weight M_w_ = 3158 g/mol, while the reduced Pine-MR measures a M_n_ of 1657 g/mol and a M_w_ of 2897 g/mol. This confirms that the reduction reaction moderately altered the molecular weight of the lignin, which is useful information in characterizing material properties in the cured lignin resins. 

### 3.2. Lignin Resins

We performed working curve calculations in triplicate using a window pane method outlined in our previous study [30]. Figure 3 shows the resulting working curves and Table 2 shows the calculated fundamental resin parameters along with the dosage required to print 50 µm layer height with a 62 mW laser at 0.09 mm scan line spacing. We observed a slight improvement in cure properties of the Pine-A resin over Pine-M, which is reflected as a decrease in the calculated dosage per layer. The windowpane technique evaluates cure reactivity as well as light penetration; hence, it accounted for the high reactivity of acrylate groups during free radical polymerization which as a result exhibited improved cure properties over methacrylate groups despite a negligible change in UV–Vis absorbance. A decrease in critical cure dosage and an increase in light penetration in the Pine-R samples resulted in a significant change in the per-layer print dosage. Increasing values for light penetration correspond well to UV absorption data in all resins tested. Smaller critical cure dosages suggest improved reactivity in lignin resins. This could indicate that changes occurring in lignin structure assisted in the formation of free radicals, potentially through the cleavage of aryl ether bonds [44], such as conventional photoinitiators [45]. These changes moved the required energy density of Pine-MR resin in line with the target for printing based on commercial resins.

We also measured the viscosity of all lignin resin samples, the results of which are shown in Figure 4. As expected, viscosity increases for all lignin resins compared to the base resin due to the introduction of the large lignin molecules. The Pine-A resins appears to shear thin at lower shear rates before stabilizing at around 0.7 Pa s, which could be explained by poor incorporation between the modified lignin and the resin base components. All other resins exhibit Newtonian behavior in a range of approximately 0.4–0.6 Pa s, which we determine is an acceptable range for use in our printer. Pine-M and Pine-MR show stable viscosity measurements over a range of shear rates, indicating good dissolution in the resin base components.

### 3.3. Mechanical and Thermal Properties

We performed tensile testing of the casted resins to determine any mechanical difference imparted by the lignin modifications. The results of these tests are shown in Figure 5. We note an increase in modulus and tensile strength among all lignin samples compared to the base resin. We attribute this to the increased crosslink density imparted by the multifunctional lignin-acrylates, which results in increased stiffness and strength but decreased ductility. Accordingly, Pine-MR is the stiffest and strongest casted resin, since it contains additional –OH groups from treatment with NaBH_4_ and, as a result, the most acylated sites.

We also noted that Pine-A is stronger and stiffer than Pine-M. The UV–Vis and Ecrit-Dp data from above indicated that Pine-A is slightly more reactive than Pine-M, despite showing little improvement in UV–Vis transparency. Although the reason for this difference is unclear, we attribute the increases in stiffness and strength to improved crosslinking efficiency due to higher reactivity of acrylate when compared to methacrylate. 

We performed thermogravimetric analysis (TGA) on the cured resins to evaluate changes in thermal behavior. The mass percent and derivative mass percent are shown in Figure 6. Although lignin has been shown previously to increase thermal stability in polymer composites [46], the lignin resins we report here decreased in the peak and onset differential decomposition temperatures compared to the base resin, particularly in Pine-A and Pine-MR resins. We attribute this to the labile C–O bonds introduced by acylation of lignin that bind it to the polymer network [47]. The derivative mass percent curves shown in Figure 6 (bottom) are unsurprisingly complex given the multicomponent polyurethane acrylate base resin. Random scission and depropagation reactions that have been observed in acrylic polymers [48] and reformation of di-isocyanates and di-alcohols in polyurethanes [49,50] could contribute to the complexity of the decomposition behaviors of these cured resins.

### 3.4. 3D Printing with Lignin Resins

We printed objects using Pine-MR lignin resin and a Formlabs Form 1+ to qualitatively assess the printing of these resins. We generated custom print settings to provide the dosage per layer calculated by the working curve calculations above. Macro images of these prints compared to the base resin are shown in Figure 7. The lignin prints exhibited rough faces and edges in contrast to the base resin print, which indicates reduced x-y dimensional accuracy. Excess resin cured along the faces between the letters of the logo in the lignin resin print, and the top layer had a smoothing out of the edges that is not present in the base resin print. Likewise, artifacts can be seen on some of the lignin print faces, indicating inconsistent curing and shrinking in some layers, which may arise from the inherent variability in the oligomeric feedstock. Future research will look to address these challenges.

Visually, the lignin prints are dark brown-amber and are partially translucent, although this property was not quantified. This result is expected based on the UV–Vis absorption data, which showed opacity primarily in UV wavelengths. Due to this, it is possible to incorporate pigments into the resin formulation to achieve a limited range of colors, depending on the UV absorption of the pigments (Figure 8). This is important for the development of lignin polymers as the natural brown color is viewed unfavorably and considered a disadvantage to using lignin, especially in photopolymers where color additives can have significant effects on processing.

## 4. Conclusions

In this study, we evaluate the efficacy of different organosolv pine lignin modification techniques in reducing the UV absorption and improving cure properties of photo-active acrylate resins designed for SLA additive manufacturing. Fundamental resin parameters showed an increase in light penetration and a decrease in required cure dosage when comparing methacrylic acylation modification to acrylic modification. The cure properties were improved further by a sodium borohydride reduction reaction prior to methacrylic acylation. Lignin resins at 5 wt.% loading showed an increase in Young’s modulus and ultimate tensile strength in all cases, with the highest increase for the reduced and methacrylated organosolv pine lignin. However, thermal stability was lowered compared to the base resin. Building on this study, we will continue to investigate the effect of reduction and subsequent acrylic acylation of lignin for improved cure properties as well as mechanical performance. Future work will also determine the lignin loading percent increases allowed by these modification techniques, evaluate the potential of pigmentation applied to lignin resins, and assess and improve the print quality. Furthermore, to improve the accessibility of lignin-based SLA resins, we will subject lignosulfonate and kraft lignins, the two most abundant types of technical lignins, to reduction and acylation modifications.

## Data Availability

The data provided in the plots and tables are complete and accurate.

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
