# Peer review of "Improving UV Curing in Organosolv Lignin-Containing Photopolymers for Stereolithography by Reduction and Acylation"

_polymers, 2021, doi:10.3390/polym13203473_

Round 1

Reviewer 1 Report

The work presented in this manuscript is novel, interesting and well-suited for publication in polymers. It reports on the chemical modification of lignin to reduce UV absorption and their subsequent incorporation into photopolymers for stereolithography 3D printing. The work is very well done but is lacking in a few minor points which should be addressed before this publication can be considered complete.

  1. The introduction requires more review on other types of materials used in stereolithography 3D printing. Some discussion of the compatibility of different SLA base resins with the authors’ modified lignin will also be interesting to include either in the introduction or elsewhere. There are some interesting examples in this review you can consider citing as a reference (https://doi.org/10.3390/polym13183101).
  2. There are two figures labelled as ‘figure 1’. Please rectify.
  3. In table 2, it would be good to briefly explain the abbreviations directly under table footnotes, aside from their explanation under the materials and methods section.
  4. The paper is lacking in more 3D printed structures using lignin-incorporated photopolymers. The printed structure in figure 7 contains defects. It is good for the authors to acknowledge problems with the new photopolymers however there should also be more photos of successful 3D prints. The current print also poorly illustrates the translucency of the new printing material, better photos and/or prints should be provided.

Author Response

The work presented in this manuscript is novel, interesting, and well-suited for publication in polymers. It reports on the chemical modification of lignin to reduce UV absorption and their subsequent incorporation into photopolymers for stereolithography 3D printing. The work is very well done but is lacking in a few minor points which should be addressed before this publication can be considered complete.

  • The introduction requires more review on other types of materials used in stereolithography 3D printing. Some discussion of the compatibility of different SLA base resins with the authors’ modified lignin will also be interesting to include either in the introduction or elsewhere. There are some interesting examples in this review you can consider citing as a reference (https://doi.org/10.3390/polym13183101).
    1. We have included a brief review of different biobased materials used in SLA resin formulations as per the reviewer’s suggestion, both in the introduction (Lines 70-82) and discussion (Lines 206-209). We have cited the recommended article and included additional, relevant publications as listed below:
      1. Tan, L.J.; Zhu, W.; Zhou, K. Recent progress on polymer materials for additive manufacturing. Adv Funct Mater 2020, 30, 2003062 DOI: 10.1002/adfm.202003062.
      2. Zakeri, S.; Vippola, M.; Levänen, E. A comprehensive review of the photopolymerization of ceramic resins used in stereolithography. Addit Manuf 2020, 35, 101177 DOI: 10.1016/j.addma.2020.101177.
  • Wagner, K. S. Investigate methods to increase the usefulness of stereolithography 3D printed objects by adding carbon nanotubes to photo-curable resins. University of Minnesota, Duluth, MN (United States): 2014; pp 1-12 https://hdl.handle.net/11299/187417.
  1. Feng, X.; Yang, Z.; Chmely, S.; Wang, Q.; Wang, S.; Xie, Y. Lignin-coated cellulose nanocrystal filled methacrylate composites prepared via 3D stereolithography printing: Mechanical reinforcement and thermal stabilization. Carbohydr Polym 2017, 169, 272-281 DOI: 10.1016/j.carbpol.2017.04.001.
  2. Ebers, L.-S.; Arya, A.; Bowland, C.C.; Glasser, W.G.; Chmely, S.C.; Naskar, A.K.; Laborie, M.-P. 3D printing of lignin: Challenges, opportunities and roads onward. Biopolymers 2021, 112, e23431 DOI: 10.1002/bip.23431.
  3. Zhang, S.; Li, M.; Hao, N.; Ragauskas, A.J. Stereolithography 3D printing of lignin-reinforced composites with enhanced mechanical properties. ACS Omega 2019, 4, 20197-20204 DOI: 10.1021/acsomega.9b02455.
  • Ibrahim, F.; Mohan, D.; Sajab, M.S.; Bakarudin, S.B.; Kaco, H. Evaluation of the compatibility of organosolv lignin-graphene nanoplatelets with photo-curable polyurethane in stereolithography 3D printing. Polymers 2019, 11, 1544 DOI: 10.3390/polym11101544.
  • Maines, E.M.; Porwal, M.K.; Ellison, C.J.; Reineke, T.M. Sustainable advances in SLA/DLP 3D printing materials and processes. Green Chem 2021, 23, 6863-6897 DOI: 10.1039/D1GC01489G.
  1. As shown in the above-cited publications, methacrylate and acrylate bases are most commonly used in SLA. Hence, the lignin resins produced in this work will be compatible with most of the common types of commercial SLA resins. An explanation to this effect has been included in the text (Lines 226-228).
  • There are two figures labelled as ‘figure 1’. Please rectify.
    1. Thank you for the suggestion, we have renamed Figure 1 to Scheme 1 and noted it in the text (Line 224).
  • In table 2, it would be good to briefly explain the abbreviations directly under table footnotes, aside from their explanation under the materials and methods section.
    1. We have explained the abbreviations in the figure title, as per the suggestion.
  • The paper is lacking in more 3D printed structures using lignin-incorporated photopolymers. The printed structure in figure 7 contains defects. It is good for the authors to acknowledge problems with the new photopolymers however there should also be more photos of successful 3D prints. The current print also poorly illustrates the translucency of the new printing material, better photos and/or prints should be provided.
    1. The reviewer has put forth a plausible concern, but despite the minor “defects” highlighted in Figure 7 the print quality was exceptional considering that our SLA resin contained appreciable quantities of lignin. These “minor defects” were only highlighted from an academic point-of-view, such that in future we could progress towards fine tuning the SLA printer settings as well as the resin formulation.

Reviewer 2 Report

This paper describes modification of lignin for use in UV-curing applications.  The authors carried out reduction and acylation of lignin and found that sodium borohydride reduction reaction prior to acylation showed an increase in light penetration and a decrease in required cure dosage.  And resins made with the modified lignin showed increased stiffness and strength with lower thermal stability.  I think the experiments were carefully done and the results will give useful information in this research field.  I would like to accept this manuscript in Polymers.

May I have some comments.

- What stands for S-OH, G-OH, and H-OH in Table 1?

- Although molecular weight obtained from GPC is sometimes incorrect, Mn = 1279 or 520 is too low.

- Both reduction and acylation of lignin did not give large effect on UV transparency (Figure 2).  However, Pine MR showed remarkable improvement.  Why?

- I'm curious about UV-absorption of, for example, acetophenone and 1-phenylethanol.

- Probably, many readers will be interested in degree of acylation.

- pin lignin --> pine lignin (in Conclusions, line 328)

Author Response

Response to Reviewer #2

- What stands for S-OH, G-OH, and H-OH in Table 1?

Thank you for the suggestion, we have labeled these abbreviations in the figure title.

- Although molecular weight obtained from GPC is sometimes incorrect, Mn = 1279 or 520 is too low.

The reviewer’s concern is duly noted. The problem is related to improper integration of the GPC chromatograms and false suppression of molecular weight data. Accordingly, we repeated the molecular weight data analysis and determined the new Mn for Pine-MR was 1657 g/mol. Proper correction was provided in Line 284.

The reason for such lower molecular mass is because we used organosolv lignin from pine wood, which naturally has a low Mn of 900 g/mol. This value is in line with previous reports by Ragauskas et al. (Characterization and analysis of the molecular weight of lignin for biorefining studies, 2014. DOI: https://doi.org/10.1002/bbb.1500), where organosolv lignins were shown to exhibit Mn in the range of 700 to 1500 g/mol.

- Both reduction and acylation of lignin did not give large effect on UV transparency (Figure 2).  However, Pine MR showed remarkable improvement.  Why?

As given in Line 264, the Pine-MR sample showed 60% reduction of absorbance in the SLA printing range of 405 nm when compared to the unmodified lignin (Figure 2 inset). This is a substantial reduction. Hence, Pine-MR displayed remarkable UV curing capacity (Figure 3, Table 2) and superior mechanical performance (Figure 5).

In addition to the proposed reduction in chromophores in Pine-MR (Lines 266-274) and increase in miscibility with the base resin formulation (as a result of increase in degree of modification shown in Table 1), there may be other lesser-known factors that affect the UV-cure properties of lignin resins and the resulting mechanical performance. We are currently investigating these factors and may follow-up with another publication.

- I'm curious about UV-absorption of, for example, acetophenone and 1-phenylethanol.

Based on the National Institute of Standards and Technology (U.S. Department of Commerce), the lmax for acetophenone and 1-phenylethanol are 187 nm and 240 nm, respectively. Kindly note that our lignin sample is an oligomer and has a broader range of absorption as shown in Figure 2.

- Probably, many readers will be interested in degree of acylation.

Thank you for the kind suggestion. But in Table 1, we have provided the direct quantification of lignin –OH groups measured using 31P NMR spectroscopy, which is also an indirect measurement of the degree of acylation (or the amount of acrylate/methacrylate groups added to lignin). We have included an additional column in Table 1 which summarizes the percent change in lignin -OH groups resulting from all chemical modifications. We hope this satisfies the reviewer.

- pin lignin --> pine lignin (in Conclusions, line 328)

Thank you, we have corrected this error (please see Line 394).

Reviewer 3 Report

The conclusions are described very generally and briefly. Future planned works should be described in more greater detail. This information may be of interest to the readers of the article and encourage them to read the next publications of these authors.

Author Response

Response to Reviewer #3

The conclusions are described very generally and briefly. Future planned works should be described in greater detail. This information may be of interest to the readers of the article and encourage them to read the next publications of these authors.

Thank you for the kind suggestion. We have added detailed future steps in Lines 403-405 and 407-409 in conclusion. We hope this satisfies the reviewer.